# What to Expect of Feeding Abilities and Nutritional Aspects in Achondroplasia Patients: A Narrative Review

**DOI:** 10.3390/genes14010199

**Published:** 2023-01-12

**Authors:** Elisabetta Sforza, Gaia Margiotta, Valentina Giorgio, Domenico Limongelli, Francesco Proli, Eliza Maria Kuczynska, Chiara Leoni, Cristina De Rose, Valentina Trevisan, Domenico Marco Romeo, Rosalinda Calandrelli, Eugenio De Corso, Luca Massimi, Osvaldo Palmacci, Donato Rigante, Giuseppe Zampino, Roberta Onesimo

**Affiliations:** 1Università Cattolica del Sacro Cuore, 00168 Roma, Italy; 2Center for Rare Diseases and Birth Defects, Department of Woman and Child Health and Public Health, Fondazione Policlinico Universitario A. Gemelli IRCCS, 00168 Roma, Italy; 3Pediatric Neurology Unit, Fondazione Policlinico Universitario A. Gemelli IRCCS, 00168 Roma, Italy; 4Pediatric Neurology Unit, Università Cattolica del Sacro Cuore, 00168 Roma, Italy; 5Institute of Radiology, Fondazione Policlinico Universitario A. Gemelli IRCCS, 00168 Roma, Italy; 6Otorhinolaryngology Unit, Fondazione Policlinico Universitario A. Gemelli IRCCS, 00168 Roma, Italy; 7Pediatric Neurosurgery, Fondazione Policlinico Universitario A. Gemelli IRCCS, 00168 Roma, Italy; 8Department of Neuroscience, Università Cattolica del Sacro Cuore, 00168 Roma, Italy; 9Department of Orthopedics and Traumatology, Fondazione Policlinico Universitario A. Gemelli IRCCS—Università Cattolica del Sacro Cuore, 00168 Roma, Italy

**Keywords:** achondroplasia, feeding, FGFR3, nutrition, obesity, review, skeletal dysplasia

## Abstract

Achondroplasia is an autosomal dominant genetic disease representing the most common form of human skeletal dysplasia: almost all individuals with achondroplasia have identifiable mutations in the fibroblast growth factor receptor type 3 (*FGFR3*) gene. The cardinal features of this condition and its inheritance have been well-established, but the occurrence of feeding and nutritional complications has received little prominence. In infancy, the presence of floppiness and neurological injury due to foramen magnum stenosis may impair the feeding function of a newborn with achondroplasia. Along with growth, the optimal development of feeding skills may be affected by variable interactions between midface hypoplasia, sleep apnea disturbance, and structural anomalies. Anterior open bite, prognathic mandible, retrognathic maxilla, and relative macroglossia may adversely impact masticatory and respiratory functions. Independence during mealtimes in achondroplasia is usually achieved later than peers. Early supervision of nutritional intake should proceed into adolescence and adulthood because of the increased risk of obesity and respiratory problems and their resulting sequelae. Due to the multisystem involvement, oral motor dysfunction, nutrition, and gastrointestinal issues require special attention and personalized management to facilitate optimal outcomes, especially because of the novel therapeutic options in achondroplasia, which could alter the progression of this rare disease.

## 1. Introduction

Achondroplasia (ACH) is an autosomal dominant genetic disease representing the most common form of human skeletal dysplasia [1,2] and the most readily recognizable dwarfing disorders [3], accounting for 90% of cases of disproportionate short stature [4]. It has an estimated prevalence in the USA of 0.36–0.6 per 10,000 births [5], affecting at least 250,000 people worldwide [6]. 

Genetically, approximately 99% of ACH cases are explained by mainly two variants in the *FGFR3* gene: c.1138G > A accounting for 98% of cases, while the remaining 1% of the patients carry the c.1138G > C variant. Both pathogenic variants, mapped on chromosome 4, result in the same glycine to arginine substitution in the *FGFR3* protein (p.Gly380Arg) [3,7,8,9,10,11]. These mutations activate the *FGFR3* receptor in the inhibition of chondrocyte proliferation with subsequent growth restriction and impaired endochondral bone formation [12]. De novo mutations account for most cases (80%) [13], and homozygosity occurrence has been reported seldomly to be compatible with life [14,15]. Its diagnosis is usually suspected prenatally by routine ultrasonographic investigations in the case of the detection of shortened limbs and confirmed by molecular testing [16]. 

Striking clinical features are macrocephaly, marked limb shortening, exaggerated lumbar lordosis, genu varum, brachydactyly, ligamentous laxity, hypotonia, and hands with a trident appearance. Face appearance is characterized by frontal bossing and mid-face hypoplasia [17]. 

Children with ACH have cognitive abilities within normal limits [18], although they can show specific difficulties in executive functioning and attention as well as mathematics and verbal intellect. Moreover, a tendency toward increased difficulties in social and emotional functioning has also been reported [19].

During infancy, special concerns arise from the cranio-cervical junction, as its constriction places children at a higher risk of severe respiratory problems [3]. Additionally, throughout the entire life span, excessive weight gain is frequently observed and constitutes an important source of morbidity [20]. Especially in adulthood, obesity can exacerbate health risks associated with lumbar stenosis, joint complaint, and cardiovascular diseases [21]. 

Although the most common disease aspects of the condition have been extensively described, little information is available on the feeding skills, gastrointestinal, and nutritional issues. 

The International Achondroplasia Consensus Statement Group underlines the importance of the regular monitoring of the condition in all of its aspects to facilitate the enhancement and standardization of care at a time when new treatment options are emerging [22].

This is the first review of the knowledge that aims to describe the evolution of oral motor abilities, nutrition, and gastrointestinal findings in ACH, and their respective treatment strategies.

The following article is in accordance with the Narrative Review Reporting Checklist [23] (Appendix A; Table 1).

## 2. Feeding

Although feeding and swallowing difficulties rarely lead to the exclusion of oral feeding in favor of alternative feeding, they should be considered in the comprehensive evaluation of the patient with ACH and managed accordingly, as further explained. Early in life, feeding issues may arise due to multiple factors including syndrome-specific oro-facial features, respiratory problems, neurological and neurodevelopmental issues, gastro-esophageal reflux, or some combination of these. Herein, we explore in detail the different causing factors and report the proposed management strategies (Table 2).

### 2.1. Oral Findings Affecting Feeding

Characteristic clinical features of ACH include muscular hypotonia during infancy and midface hypoplasia. 

With regard to floppiness, the medical literature reports that almost all infants with ACH exhibit variable levels of hypotonia during infancy. This common finding is a contributing factor to delayed motor development [37], consistently reported in multiple studies [6,38,39,40]. Hypotonia also negatively influences the optimal development of oral-motor skills by reducing the strength of orofacial muscles. Consequently, jaw movements, lip tightness, tongue positioning, sucking/swallowing/respiration pattern, and general feeding behavior are compromised. In addition, hypotonia of the oral musculature along with midface hypoplasia and enlarged tonsils and/or adenoids is one of the leading causes of the narrowing or even obstruction of the upper airways [41,42], which occasionally requires endotracheal tube placement [43]. The presence of relative macroglossia further contributes to airway obstruction as the tongue encroaches upon and intermittently obstructs the retroglossal airway [44], leading to swallowing difficulties.

Upon closer look, in patients with ACH, muscular weakness can be associated with a tongue thrust swallowing pattern [45]. Normally, between 2 and 4 years, the infantile swallowing pattern changes gradually into a mature swallowing pattern that implies the positioning of the tongue held high on the palate behind the maxillary incisors [46]. In the tongue thrust pattern, instead, the tongue pushes against or between the teeth, leading to open bite and protruded teeth [47]. 

Moreover, striking ACH facial manifestations, in addition to mouth breathing predisposition, also interfere with the chewing ability [48].

Concerning the typical oro-cranio-facial findings in ACH children, they include macrocephaly, prominent forehead and frontal bossing, and underdevelopment of the cartilaginous bones of the face. These characteristics result in midface hypoplasia, collapsed midface, and an elongated lower face and concave profile [32]. In turn, oral findings include posterior crossbite, anterior open bite, prognathic mandible, and retrognathic maxilla with reverse overjet and high-arched palate [49,50]. In this specific genetic condition, maxillary hypoplasia typically results in an Angle class III malocclusion with anterior open bite [32]. To note, the deviation from normal mandible growth can strongly affect the masticatory and respiratory functions [51]. 

Finally, the development of adequate chewing ability in children can be affected by a delayed eruption of teeth and oligodontia due to altered bone growth [52,53,54]. In adults, instead, a deterioration of chewing function can be caused by progressive teeth loss due to cartilage formation impairment, as observed by Swathi et al. in a 50 year-old-man with partial edentulism [50], and to dental mobility, as observed by Chawla et al. in a 31-year-old female patient with severe periodontal disease [55].

### 2.2. Respiratory Findings Affecting Feeding

Many infants and children with ACH present respiratory complications due to several factors, namely, upper airway obstruction, neurological dysfunction, and rib cage deformity [56]. This latter feature is one of the main causes of respiratory decompensation in ACH [57]. In normal infants, the thorax is circular at birth with minimal difference between the thoracic width and depth. With growth, the transverse posterior diameter becomes larger than the anterior–posterior one, giving an elliptical appearance to the thorax. At the end of growth, the thoracic width and depth represent approximately 30% and 20% of sitting height, respectively [58]. Conversely, because of irregular rib development, ACH infants have a small thorax with short flared ribs, the thorax is frequently bell-shaped, and has reduced anterior–posterior diameter [57]. 

Infants with decreased total lung capacity commonly have a rapid and shallow respiratory pattern because of increased respiratory frequency and diminished tidal volume: these infants often exhibit increased work in breathing, which is a distinctive feature of syndromes with short rib dysplasia and tachypnea [59]. Of note, persistent marked tachypnea can cause secondary feeding difficulties [3] and increase the risk for aspiration [60]. Furthermore, Dessoffy et al. first established an estimated frequency of 5.5% for airway malacia in the ACH population, which is considerably higher than the cumulative frequency in average statured children (from 0.5% to 1.5%) [61]. The presence of laryngomalacia or trachea-bronchomalacia negatively influences swallowing and feeding functions [62], commonly causing regurgitation, choking, and slow feedings [26,63]. Empirically, laryngomalacia involves a collapse of the supraglottic structure during inspiration: it contributes to a failure of sucking, swallowing and breathing coordinated pattern, and airway protection. Therefore, laryngeal penetration and aspiration are common findings [31]. Meanwhile, trachea-bronchomalacia, due to the softening of the tracheobronchial tree, can cause a wide range of respiratory and feeding problems including dysphagia, cough, and cyanosis [64,65].

### 2.3. Neurological Findings Affecting Feeding

ACH is characterized by impaired enchondral ossification, which gives rise to neurologic abnormalities including foramen magnum stenosis (FMS). FMS may be common in ACH, and in about 10–20% of infants leads to cervico-medullary myelopathy and a range of symptoms including poor suck, poor weight gain, and weakness [27].

Therefore, swallowing difficulties, in addition to lower cranial nerve palsies, hyperreflexia, generalized hypotonia, weakness, and clonus, can suggest a cervical myelopathy [4,19]. 

Another neurological manifestation in infants with ACH is hydrocephalus, occurring in 15–50% of patients [66]. Hydrocephalus is due to increased intracranial venous pressure secondary to stenosis of the jugular foramina [4] and, if occurring, may further contribute to poor feeding [6].

In the study by Ireland et al. [6], the authors first described the development of feeding skills in 20 Australasian children diagnosed with ACH. Using a retrospective questionnaire covering the progression in the introduction of food texture, the authors found an overall adequate sucking ability with a preference for exclusive breastfeeding over bottle-feeding in half cases. Moreover, the timing of the introduction of semi-solid food textures (median: 5 months) in children with ACH was in line with those in the general population. 

In contrast, self-feeding with a spoon (median: 20.5 months), cup drinking (median: 20 months), and finger-feeding skill (median: 15 months) attainment faced a delay if compared with the general population. 

In this regard, Morris and Klein highlighted how eating efficiently by mouth relies largely on the steadiness or stability of the head, neck, and trunk. The absence of postural control and alignment may preclude the foundational support and stability essential for the refined motor skill actions required during feeding [67]. Difficulty in attaining the sitting position and the placement in a reclined position, necessary to minimize further kyphotic development, implies the lack of postural alignment required for effective chewing and swallowing. The rhizomelic limb shortening and lack of full elbow extension further delay the development of self-feeding skills and increase reliance on caregivers compared to peers [38], Figure 1.

### 2.4. Feeding Management

The primary goals of the management of feeding-related issues in the ACH population are to prevent complications and introduce personalized treatments in a timely manner with constant involvement of the family in decision-making. As previously outlined, midface hypoplasia is universal in the ACH population as the result of impaired endochondral bone formation and normal membranous ossification [68]. Therefore, maxillary hypoplasia, relative mandibular prognathism, and class III malocclusion accompanied by oral muscle dysfunction are consistent features [69]. The American Academy of Pediatrics recommends a review of orthodontic problems in ACH after 5 years of age [70]. The main goal of orthodontics is primarily to enhance maxillary and restrict mandibular growth. As proposed by Pineau et al., there is a need in this specific population to correct the anterior crossbite and open bite, improve the skeletal class III jaw-base relationship, create proper overjet and overbite, and establish an acceptable occlusion with a functional class I occlusion [32]. Correction of malocclusion with orthodontic strategies and myofunctional therapy go hand in hand, with the latter therapy lasting throughout the orthodontic treatment [24]. Generally, myofunctional strategies include exercises involving the cervical and facial muscles aimed at improving proprioception, tone, and mobility. The myofunctional approach, in turn, results in neuromuscular re-education of the muscles involved in swallowing, tongue motion, oral breathing as well as the rest posture of the tongue, lips, and cheeks [47]. In this specific context, myofunctional strategies are intended to stop the tongue-thrusting habits, optimize glossal muscle tone, and correct its positioning and functioning. A further management strategy consisting of personalized craniofacial surgery may also be needed in some selected cases [32]. Usually, although with increasing age there is a spontaneous improvement in oral motor function, a lack of autonomy achievement during feeding may result in greater caregiver dependence. Therefore, multidisciplinary assessment should focus on areas of vulnerability and appropriately promote the support of the entire family in all aspects of the patients’ daily life. 

Due to the anatomical features of ACH infants, management guidelines recently developed by the European Achondroplasia Forum (EAF) and the American Academy of Pediatrics (AAP) indicate considerations for early infant handling including breast- and bottle-feeding positioning. In everyday practice, when breastfeeding, it is recommended to support the infant head and neck and use a firm support for kyphosis. If bottle feeding, it is recommended to support the infant’s back using a pillow with a firm hand on the lower back. With growth, there is still the need to support the children’s kyphosis [25,71].

The presence of restrictive pulmonary disease may cause tachypnea, consequent feeding difficulties, and failure to thrive. Therefore, the management of these latter aspects cannot be separated from the assessment of respiratory functioning. Polysomnography and daytime spot oximetry during active alert time and, particularly during feedings, for example, may be helpful [3]. 

As previously described, tracheo and bronchomalacia are commonly present after the neonatal period with airway symptoms including cough, stridor, airway obstruction, frequent infections, and wheezing [26]. It is recommended that a multidisciplinary team of clinicians is involved in the counseling of families and caregivers to better define the most appropriate management of airway malacia including supraglottoplasty or more conservative options [31]. Concerning the management of neurological findings, the cranio-cervical junction constriction, of which poor feeding is an indirect symptom, is a major concern in ACH patients. Indeed, the comprehensive history and physical exam for foramen magnum stenosis should include the evaluation of weak suck and feeding difficulty [27]. Following the American Academy of Pediatrics guidelines on health supervision for children with ACH, magnetic resonance imaging (MRI) of the cervical–cranium region is strongly advised to screen all infants with ACH [69]. International experts recommend a neurological evaluation from infancy [recommendation #34] [22] with special attention also on symptomatic hydrocephalus. Since birth, all children with ACH should have routine circumference measurements plotted on ACH-specific head circumference charts [3,8] and further neurosurgical evaluation if rapid growth and other clinical or neuroradiological signs are observed [22]. 

## 3. Gastrointestinal Issues

Despite gastrointestinal issues including gastroesophageal reflux (GER) being able to further complicate the picture in infants with ACH [3,72], they are scarcely reported, and gastroenterologists are rarely included in inter-disciplinary teams treating skeletal dysplasia patients [29]. 

With regard to pediatric age, Tasker et al. reported five cases with severe or moderately severe GER, as shown by barium swallow studies and lower esophageal pH probe monitoring. The authors observed a worsening in respiratory compliance with delayed lung growth in the presence of GER [72]. They also reported that severe GER may have a role in the so-called “achondroplasia respiratory difficulty syndrome” [30], both in leading and in being led by lung injury and hyperinflation. In particular, GER could lead to hyperinflation and small airway disease by three mechanisms of aspiration of gastric contents, which differ based on the depth of the airways reaching the lower respiratory tract, as far as the pharynx, causing bronchoconstriction, and limited to the lower esophagus, causing symptoms either by increasing bronchial reactivity or by bronchoconstriction secondary to a vagal reflex [73]. Therefore, because severe GER can lead to worsening of respiratory compliance in people with ACH, early recognition is mandatory using the clinical history and possible instrumental investigation through esophageal pH monitoring [72].

In the study by LoTurco et al., the frequency and characteristics of gastrointestinal symptoms in adults with skeletal dysplasia including ACH were obtained through the administration of the Gastrointestinal Symptom Rating Scale (GSRS). Scores in patients with ACH were higher for indigestion, constipation, diarrhea, and lower for abdominal pain and reflux when compared with the scores obtained by the general population, with the diarrhea score differing significantly (*p* = 0.034) [29]. 

In 2017, Kylat et al. reported a unique case of biliary atresia in a newborn with ACH [74]. In the same year, a 62-year-old patient with achalasia and Serrated Polyposis syndrome was first reported by Umar et al. [28]. Due to the *FGFR3* gene mutation involved in the regulation and proliferation within the colonic epithelium, further attention should be paid to colorectal cancer surveillance and detection [28].

## 4. Nutritional Aspects and Obesity

### 4.1. Nutritional Findings and Obesity

ACH is known to be associated with an increased risk of obesity [4] and a major predisposition to abdominal obesity since early childhood, though the causes are still not fully understood [1,71,75]. It has been recognized as a major health problem in adults with ACH and it can worsen complications such as lumbar spinal stenosis, joint pain, or sleep apnea [76,77,78]. In their recent case-control study, Fredwall et al. detected a higher prevalence of obesity with a body mass index (BMI) ≥ 30 kg/m^2^ among 49 Norwegian adults with ACH, specifically 67% (*n* = 33/49) [79]. Another recent study in a 33 Norwegian adult population with ACH concerning the resting energy expenditure (REE) revealed lower daily REEs compared to the controls [80]. These data are consistent with the previously reported findings by Owen et al. in the adult American and Canadian ACH populations [81]. 

Nowadays, the most accredited hypothesis is that the predisposition to obesity and its higher prevalence in ACH results from an energy disbalance due to excessive calorific intake and lack of physical activity [3]. Due to their physical condition, the exclusion from commonly practiced sports limits energy expenditure and increases the sedentary lifestyle of persons with ACH. Furthermore, Saint-Laurent et al. highlighted the presence of an abnormally increased appetite among children with ACH [82], as also previously noted in mice models [1]. Through the animal model, the role of the *FGFR3* gene and its potential nutrigenetics effect have been studied [1].

Focusing on ACH metabolic characteristics, as reported in the recent review of Saint-Laurent et al. [82], the atypical visceral obesity development in ACH is not associated with a diabetic profile, but rather with low insulin and glucose levels with a tendency to lower fasting glycemia to increase [1,82,83]. Moreover, a preference for fat oxidation over carbohydrate oxidation has been observed.

### 4.2. Management of Obesity and Nutritional Aspects

The assessment of nutritional status in ACH patients is an important part of their health monitoring. The method of choice to analyze body composition in people with ACH is dual-energy X-ray absorptiometry (DEXA) [84]. DEXA was initially developed for the diagnosis of osteoporosis, and successively used to assess the fat and fat-free mass following the use of whole-body scans [85]. In ACH patients, DEXA is recommended to assess the body adipose percentage rather than bone density [86] as it under predicts the mineral density of long bones [84]. 

To better understand how to nutritionally follow-up ACH children, different specific growth and BMI charts have been proposed. Height or length should be measured routinely and compared with growth references available from European, Japanese, American, Argentine, Australian and Egyptian children and adolescents with ACH [86,87,88,89,90,91,92,93]. However, standards to evaluate obesity available in children and adults with ACH are still lacking [93].

Weight control management is often associated with interventional clinical approaches aimed at growth enhancement. Therefore, prompt dietary management of children with ACH can optimize their chance for appropriate body proportion alongside possible lengthening interventions including the recently approved pharmacologic approach with vosoritide [86]. It is valuable to mention the importance of regular anthropometric measurements during limb lengthening interventions. Orthopedic procedures are intended to increase the leg length and height and ameliorate the trunk-lower limb proportion [94], but at the same time imply a reduction in the possibility of regular physical activity with subsequent increased risk of obesity. Considering the metabolic characteristics, modifying the percentage of macronutrients with respect to the total energy value of the diet may be one of the keys to dietary intervention in ACH. Personalizing recommended proteins, carbohydrates, and fat percentages to their real needs can help to control obesity [35].

In particular, as for carbohydrates, it could be necessary to reduce or remove free sugars in favor of complex ones [33]. As for lipids, it seems to be essential to promote the consumption of monounsaturated and polyunsaturated lipids with cardioprotector effects and to reduce the consumption of saturated and trans or hydrogenated fats [95]. However, there is a lack of data regarding the ideal oral intake in infants with ACH and other dwarfism diseases or clear information about the appropriate weight gain velocity in these patients [96]. Although obesity may negatively affect the quality of life (QoL) because it predisposes to reduced mobility, breathing difficulties, and joint pain, weight loss remains challenging to obtain [22]. The incentive of physical activity including swimming will prevent not only weight increase, but also has a role in mental well-being [34].

Knowing that people with ACH and obesity must face a double stigma and social exclusion, causing stress and anxiety, and have a greater cardiovascular risk and a worsening of comorbidities such as sleep apnea, weight monitoring and education regarding healthy eating should be provided at each follow-up appointment involving the family [36]. In this context, psychological support for the patient and family through interactions with national associations is a valuable aid in improving the QoL and promoting autonomy. 

## 5. Conclusions

This review summarizes the feeding and nutrition difficulties that people with ACH can face from early infancy transitioning to adulthood and the relative treatment strategies. Feeding and nutritional aspects represent a compelling array of issues in ACH for the practitioner to consider during the entire lifespan. Management of these issues are part of an evolving context in which new pharmacological approaches may change the natural history of the condition. Oro-motor development may be delayed during childhood, while with age, issues arise concerning the BMI and the waist circumference increase. Finally, anticipatory care and proactive care should be directed at identifying and managing the issues that can have a negative impact on a patient’s life. 

## Figures and Tables

**Figure 1 genes-14-00199-f001:**
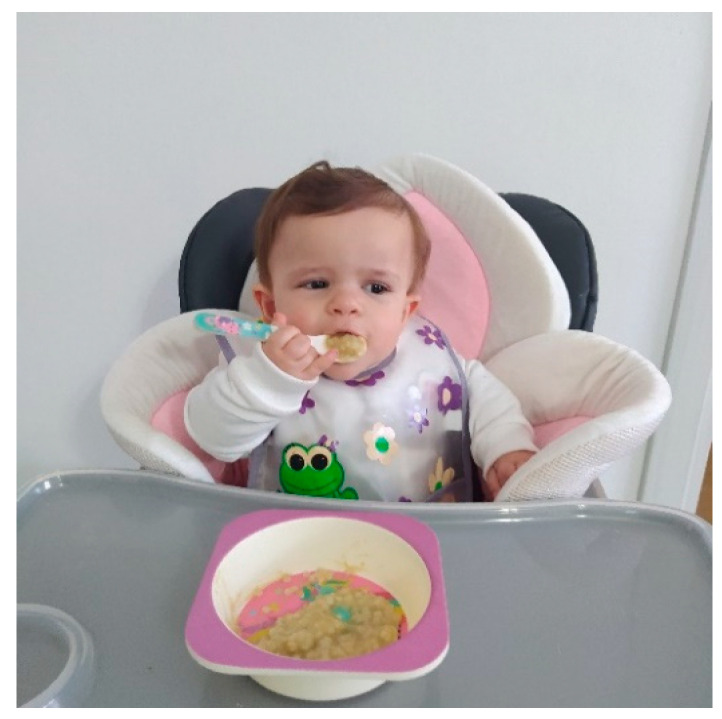
A young patient with achondroplasia during feeding. The handle used for the cutlery allows the child to feed herself despite the rhizomelic shortening of the upper limb.

**Table 1 genes-14-00199-t001:** Search strategy summary.

Items	Specification
Date of search	November 1st 2022
Databases and other sources searched	PubMed, Scopus, Cochrane Library
Combined search terms used	“achondroplasia”, “FGFR3”, “feeding”, “nutrition”, “gastroenterology”, “obesity”, “management”
Timeframe	Date unrestricted to November 2022
Inclusion and exclusion criterion	Inclusion: (I) English language; (II) Case reports, case series, retrospective cohort series, reviews, guidelines, consensus opinions; (III) Focusing on: nutrition, feeding, dysphagia, gastroenterology, obesity; (IV) Population: children and adult. Exclusion: full text unavailability
Initial results	*n* = 200
Final results	*n* = 96

**Table 2 genes-14-00199-t002:** Clinical features affecting feeding and nutrition during the lifespan and their management strategies.

Age	Clinical Features	Impact	Possible Management Strategies
Infancy	Hypotonia	Strength reduction of the orofacial muscles	Neurological consultation [22] Myofunctional therapy [24] Specific handling strategies during feeding [25]
	Airway malacia and/or Tachypnea	Respiratory and feeding difficulties	ENT consultation [3,26] Bronchopulmonary consultation Respiratory support (CPAP) [3,26] Family counselling
	Foramen magnum stenosis and/or hydrocephalus	Poor suck, poor weight gain and weakness	Neurosurgery consultation [22,27] MRI follow up [28]
Childhood	Midface hypoplasia	Obstruction of the upper airway Immaturity of the masticatory pattern	ENT consultation [3] Bronchopulmonary consultation [3] Respiratory support (CPAP) [3] Orthodontic interventions [24] Myofunctional therapy [24]
	Rhizomelic limb shortening and lack of full elbow extension	Delay in developing self-feeding skills	Child support by the caregivers Family counselling
	Gastroesophageal reflux	Worsening of respiratory compliance Worsening of QoL and mealtimes	Lifestyle change [29] Gastroenterological consultation [29] Proton pump inhibitors therapy [29]
	Constipation	Worsening of QoL and mealtimes	Toilet training [30] Pharmacological approach [30]
Adulthood	Periodontal disease	Deterioration of chewing function	Conservative treatments [31] Surgery [32]
	Obesity	Worsening of comorbidities Decrease of QoL	Promotion of healthy eating [33] Promotion of physical activity [34] Personalized diet [35] Counselling on satiety control [36]

CPAP: continuous positive airway pressure; ENT: otolaryngologist; MRI: magnetic resonance imaging; QoL: quality of life.

## Data Availability

Not applicable.

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
