# Peer review of "What to Expect of Feeding Abilities and Nutritional Aspects in Achondroplasia Patients: A Narrative Review"

_genes, 2023, doi:10.3390/genes14010199_

Round 1

Reviewer 1 Report

This is an excellent review which focuses of various organ systems affected by achondroplasia and the resulting implications on feeding and nutrition. The concepts are presented in a clear and organized manner, and appropriate citations are provided throughout. The use of English language is excellent with only minor copyediting required.

Minor comment:

1. At present, no figures are provided. The presentation of this article could benefit from the addition of 1 or 2 figures to convey important concepts the authors wish to highlight. These can be original figures or figures taken from the literature (obtaining the appropriate permission to reproduce the figure where necessary).

Reviewer 2 Report

The article "What to expect of feeding abilities and nutritional aspects in achondroplasia patients: a narrative review" is a rather interesting article on feeding habits and a number of related conditions in patients with achondroplasia. There are some comments that could improve the quality of the publication:

1. There is a paragraph in the article stating that 'Reviews, guidelines, consensus opinions, cohort and single case studies were included in this narrative review without any date limit'. It needs to be clarified which databases, key words were used, how many publications and what types of publications were included in the review, as the 97 articles in the literature list are probably not the whole amount of data on the disease.

 (2) Table 1 needs to be clarified as to the sources from which it was compiled. Also, it is currently a low resolution figure (?), needs to be reformatted and notes with abbreviations added

3. The disease is rare, this puts a strain on the nature of the research, but I would like to see results not only of potential mechanisms that may affect feeding, but also results of comparative studies - body mass index, protein levels, other manifestations of nutritional deficiency or increased feeding in patients with achondroplasia compared to different cohorts. I saw only one mention in the article on line 286

4. A number of minor corrections

- line 297 - different brackets

- line 329 - clarify what "weight/height2" means. If it is a degree, superscript is needed

- check the text again for abbreviations the first time they are used

Reviewer 3 Report

In the manuscript titled “What to expect of feeding abilities and nutritional aspects in achondroplasia patients: a narrative review”, the authors summarized feeding and nutritional difficulties in ACH patients. It is very interesting and important for readers because the management of these issues are critical for physicians and caregivers. The authors need to revise some points as follows.

In the introduction, the authors mentioned that 90% of cases carrying Gly380Arg variant in FGFR3 gene. However, they described that all individuals with ACH have identifiable mutations in FGFR3 gene in the abstract (line11-12). They need to rephrase the “All” to “Almost all” in the sentence.

The name of gene should be described as Italic font in line 12, 288, and 313.

The resolution of Table 1 is insufficient and the size of letters is too small. The authors need to correct it.

The reviewer thinks the paragraph describing other co-occurrent genetic conditions affecting feeding is unnecessary and suggests to remove this paragraphs (line 172-205).

In line 288, the description of G1138A should be removed.

In line 329-331, the authors referred #89 as an available growth references from European, Japanese, American, Argentine and Australian children and adolescent with ACH. However, there was no such data in reference #89. The authors need to refer an appropriate reference here.

Round 2

Reviewer 2 Report

Thank you to the writing team for their work, all questions have been answered and comments have also been taken into account. I recommend the manuscript for publication